# Individual Health Management (IHM) for Stress—A Randomised Controlled Trial (TALENT II Study)

**DOI:** 10.3390/healthcare13233181

**Published:** 2025-12-04

**Authors:** Dieter Melchart, Erich Wühr, Beatrice Bachmeier, Lara Isabel Jötten

**Affiliations:** 1Faculty of Applied Healthcare Sciences, Deggendorf Institute of Technology (DIT), Competence Centre for Lifestyle Medicine and Naturopathy, 93444 Bad Kötzting, Germany; erich.wuehr@th-deg.de (E.W.); lara.joetten@th-deg.de (L.I.J.); 2Institute of Pharmaceutical Biology, Goethe-University, 60438 Frankfurt Am Main, Germany; bachmeier.beatrice@gmail.com

**Keywords:** Intensive Lifestyle Intervention (ILI), Traditional Chinese Medicine (TCM), Individual Health Management (IHM), perceived stress, TALENT-II study

## Abstract

**Background/Objectives:** Stress is a pervasive modern challenge that contributes to serious health problems and affects a significant proportion of the population. This study examines whether an Individual Health Management (IHM) programme incorporating elements of traditional Chinese medicine is effective in reducing stress. **Methods:** Participants experiencing stress were enrolled in a monocentric randomised controlled trial. The intervention group received the IHM programme and was monitored for 12 months. After six months, a primary comparison was conducted with a waitlist control group, who then also received the intervention. Additional crossover analyses were employed. The main outcome measure was the change in subjective stress (as measured by the Perceived Stress Questionnaire, PSQ) from baseline after six months. Secondary outcomes included further parameters related to stress and mental load. **Results:** The intervention group achieved normal stress levels, with a 41% reduction, while the control group experienced a 9% reduction. The adjusted mean difference in PSQ scores between the two groups was −21.02 (95% CI: −27.34, −14.70), accounting for baseline values and gender. This is the ANCOVA result for the ITT population at the 6-month follow-up. Secondary outcomes also improved significantly. **Conclusions:** The IHM has multiple moderate to strong effects on mental health, resulting in clinically normal distress scores that can persist for up to one year.

## 1. Introduction

Mental health is a key factor in the success of both individuals and society. The growing incidence of mental health issues, including anxiety and depression, puts a strain on healthcare systems and diminishes individuals’ quality of life [1,2]. This results in an increased demand for care and a greater disease burden [3]. It is therefore important to provide targeted support to improve mental health and quality of life.

Psychological distress is characterised by symptoms of stress, anxiety, and depression [4]. These symptoms often occur together and are associated with an increased risk of cardiovascular disease, mental health disorders and premature death [5,6]. Experiencing distress has also been linked to negative outcomes such as neglecting physical exercise, increased alcohol consumption and smoking, disrupted sleep patterns, and adopting an unhealthy diet [7]. To date, most studies have focused on individual stress reduction techniques, such as muscle relaxation and meditation [8,9]. While these approaches are effective at reducing acute stress symptoms, they primarily target short-term physiological and psychological responses. However, stress is a multifactorial phenomenon influenced by behavioural, social and environmental determinants, which suggests that a narrow focus on isolated stress techniques may be insufficient to achieve long-term benefits [10,11,12]. Recent systematic reviews have emphasised the critical role of lifestyle factors, including physical activity, nutrition, and sleep, in mental health and stress regulation [13]. A meta-analysis of 96 randomised controlled trials found that lifestyle interventions significantly reduced symptoms of depression, anxiety and stress [14]. Based on these findings, interest in ‘lifestyle medicine’ and ‘lifestyle psychiatry’, which advocate a holistic approach emphasising that sustainable stress management requires integrating multiple domains of daily life rather than relying solely on relaxation techniques, is increasing [15,16]. This shift reflects a growing consensus that modifying lifestyle behaviours can enhance mental health and promote long-term psychological well-being. This is why there is growing academic and clinical interest in the relationship between ‘lifestyle factors’ and mental health and psychological well-being.

Physicians and other healthcare providers should be able to care for their patients—and themselves—by not only treating cases of acute or chronic illness, but also by implementing strategies to prevent mental and physical disorders. This type of intervention, also known as ‘lifestyle medicine’, is healthier, more cost-effective and can be performed by more people [17]. Numerous experts have developed health programmes to promote healthy behaviours, and health policy initiatives now call for lifestyle behaviours to be addressed [18,19]. Further research into efficacy and implementation is necessary to develop innovative and successful programmes for lifestyle-based interventions, where other evidence-based therapies can be incorporated.

For this reason, the clinical effectiveness of a tailored lifestyle self-management intervention called the Individual Health Management (IHM) programme will be tested in relation to stress reduction. In the context of weight loss, the TALENT I study demonstrated that participants who received the IHM lost around 10% of their initial weight after one year [20]. Although the basic concept of IHM (Individual Health Management) is very broad and multifaceted [21,22], the planned study will focus on stress reduction as target parameter. Data from an own pilot study showed that the IHM is a feasible concept for a web-based lifestyle intervention programme in people with perceived stress [23]. Self-management is a key element of the Lifestyle IHM programme, enabling individuals to maintain and optimise their basic physiological functions (e.g., exercise, nutrition, breathing, sleep and body temperature regulation) and their psychosocial and cognitive abilities (e.g., emotional self-regulation and social competence) in daily life. With the help of a web-based health portal, individuals can identify and critically analyse stress-inducing situations through systematic self-observation of their feelings, thoughts and behaviour. Participants learn about various behavioural changes, for example, in the area of everyday exercise through the use of pedometers and the implementation of regular Qigong exercises, as well as a variety of other self-help techniques.

This is achieved via a blended learning programme lasting at least 12 months, combining modern e-health technologies with personal support in the form of health coaching, health days, remote support and medical advice and treatment if necessary. As a local pilot project, the study is supported by AOK Bayern (the General Health Insurance Fund of Bavaria) in cooperation with the TCM (Traditional Chinese Medicine) Clinic Bad Kötzting. The aim of the presented TALENT II study is to evaluate the efficacy of the lifestyle intervention program IHM in the reduction in perceived stress in respect to a control group (waiting list) after 6 months

## 2. Materials and Methods

### 2.1. Study Design

This randomised, two-arm, controlled, monocentric, interventional clinical trial was conducted at the SINOCUR Prevention Centre in Bad Kötzting, Germany. The centre is part of a centrally coordinated health promotion network called ‘IHM Campus’. The study compared an intervention group (IHM) with a control group who were placed on a six-month waiting list for the IHM programme. All participants underwent a main observation period of six months, with the intervention group undergoing further longitudinal examinations until the end of the 12-month lifestyle programme. Additionally, patients in the waiting list group were compared with themselves after receiving the intervention, using a crossover design. The study was pre-registered with the German Clinical Trials Register Freiburg (DRKS, file number DRKS00013040, registration date 1 October 2017). Participants were required to provide written informed consent. The study protocol was submitted to the Ethics Committee of the Medical Faculty of the Technical University of Munich (TUM; ref. 278/17S on 4 August 2017) for review of ethical and legal compliance, as the TUM is responsible for the principal investigator.

### 2.2. Recruitment, Randomisation and Participants

The health insurance company AOK Bayern recruited 84,134 participants via written correspondence. Of the 2684 who completed the online health survey, 623 participants (31.9%) showed no signs of stress. Participants were included if they scored at least 3.20 on the TEDIUM measure (indicating moderate stress or pre-burnout), scored more than 41 on the Perceived Stress Questionnaire (PSQ), and had experienced stress for more than three months. Participants were excluded if they met any of the following criteria: lack of legal capacity; insufficient knowledge of German; lack of private internet access; a body mass index (BMI) of less than 17.5; the need for psychiatric or psychotherapeutic treatment; known health problems (e.g., diabetes, heart disease or liver or kidney disease); pregnancy or lactation; or participation in another clinical trial within the last six months. The TEDIUM questionnaire revealed that 878 people (45.0%) exhibited significantly elevated stress levels indicating pre-burnout, while 427 people (21.9%) exhibited burnout (TEDIUM score ≥ 4.5). Two hundred and fifty participants who met the initial study requirements and were willing to participate were invited to meet with the study physician to receive further information and undergo an eligibility assessment. Eligible participants were randomised immediately after being formally included in the study. The trial physician opened the closed envelope in strictly sequential order according to enrolment, disclosing the allocated study arm to the participant. The allocation ratio was 1:1, meaning one for the intervention group and one for the control group (waiting list). Randomisation was prepared for 150 participants (including replacement numbers). The randomisation and allocation envelopes were prepared by an independent data manager at the Institute of Medical Statistics and Epidemiology at Technical University of Munich (TUM), later Deggendorf Institute of Technology (DIT). Participants were divided into training groups of around 12 people, with recruitment taking place every six months. This resulted in longer recruitment times. The screening phase of the study was conducted between May 2017 and April 2019. The recruitment phase of the study was between 10 March 2018 and 25 March 2023.

Sample size calculation and further details of the study protocol were described by Melchart et al. [24]. Due to difficulties arising from the pandemic, the estimated sample size of 136 (α = 0.05, two-sided, power = 80%, expected dropout rate = 5%) could not be achieved. The recruitment and allocation of participants is illustrated in the CONSORT study flowchart (Figure 1).

### 2.3. Intervention

The intervention takes a blended learning approach, combining group sessions, one-to-one counselling, and access to a personalised, web-based health portal (www.viterio.de). Through the portal, participants can access personalised feedback in the form of written reports and graphs detailing their progress. This feedback includes time and mood analysis, as well as information on lifestyle changes, behavioural adjustments and emotional regulation. With the help of a visual analog scale (VAS), participants record the intensity of their feelings and moods in everyday life and are asked to name these feelings. The training content includes exercises to promote self-awareness, optimise quality of life, and improve time and sleep management. It also includes a six-minute and two-kilometre walking test, “3-1-2” Qigong relaxation techniques, and information on stress, resilience, mood regulation, and diet. Information is also provided on health behaviours related to physical activity and nutrition, as well as on setting individual goals. Qigong and special forms of Qigong are used, such as ‘healing sounds’, which are concentrated breathing and relaxation techniques used in TCM. Additional constitutional aspects of Chinese nutritional theory are taught on an individual basis around nutrition. Participants will also learn various self-help techniques, such as acupressure and meridian massage. Further details of the IHM intervention programme can be found in previous publications [21,23].

The 12-month IHM lifestyle intervention programme includes various phases and training packages, supported by a web-based health portal (viterio.de). During the first three months, participants receive intensive lifestyle counselling: three health days (induction phase) and ten weekly after-work seminars (training phase). This is followed by a nine-month maintenance phase: weekly monitoring by health coaches, remote lifestyle counselling as needed, and half-day refresher courses (see Figure 2). Successful participation is defined as attending seven of the ten after-work seminars and at least three of the four refresher days between months three and twelve.

### 2.4. Outcomes

The primary outcome measure is subjective stress, as determined by the total PSQ score [25]. This is compared between the IHM and CG groups at six months and at baseline. The secondary outcome parameters include various self-report questionnaires, such as those measuring TEDIUM scale [26], psycho-vegetative complaints, mental stress and burden (ISR; [27]), the visual analogue scale for well-being (WHO-5; [28]), vitality and self-efficacy, optimism and pessimism (SWOP; [29]), sense of coherence (13 items; [30] and life satisfaction (FLZ; [31]). During the baseline examination, sociodemographic data is documented. Adverse event occurrence is recorded at each physical examination.

### 2.5. Statistical Analysis

At six months, perceived stress (PSQ total score) was the primary outcome, and was evaluated using ANCOVA to assess intervention effects, adjusting for baseline scores and gender (5% two-sided significance level). The intention-to-treat (ITT) population included participants with baseline PSQ data. Multiple imputation by chained equations (MICE) with ten imputed datasets was used to handle missing data. Descriptive and inferential analyses were conducted on the multiply imputed datasets and pooled using Rubin’s rules. Degrees of freedom were calculated via Barnard and Rubin’s method. Effect sizes (Cohen’s d and partial eta^2^) and standard deviations were averaged across imputations. Sensitivity analyses included a per-protocol approach (complete cases only). Additional *t*-tests were conducted to assess changes within- and between-group over time. Secondary outcomes were analysed in a similar way using the ITT sample. All tests were two-tailed at α = 0.05. No correction for multiple testing was applied due to the exploratory nature of secondary outcomes. For sample sizes >30, *t*-tests were applied regardless of normality, supported by the central limit theorem [32]. Non-parametric tests were used where appropriate.

## 3. Results

### 3.1. Sociodemographic and Baseline Characteristics

A total of 77 participants were included in the study: 38 (49%) were assigned to the intervention group and 39 (51%) to the control group. The control group was 89.7% female, while the intervention group was 76.3% female. This difference was significant (*p* < 0.001). The mean age was 45.18 years (SD = 11.35) in the control group and 44.32 years (SD = 10.05) in the intervention group (*p* = 0.725). The total PSQ score at baseline was similar in both groups. However, the groups differed in terms of the ‘joy’ subscale of the PSQ (*p* = 0.015, d = 0.57). No significant differences were observed between the groups for other parameters (see Table 1).

### 3.2. Dropout and Missing Data

The actual dropout rate was 18.4% compared to the projected 5%. To address this, we performed both per-protocol and intention-to-treat analyses, with missing data handled using multiple imputation by chained equations (MICE) under the assumption of missing at random. Three participants dropped out before month 3, four by month 6, and two by month 12. By the end of the 12-month intervention, 29 of the original 38 participants had completed the study. In the control group, the dropout rate by month six was 7.7% (three out of 39 participants). Due to technical difficulties, one participant’s PSQ data was lost, resulting in 35 complete PSQ observations in the control group and 31 in the intervention group by month six. The main reason for dropping out of the study was the COVID-19 pandemic.

### 3.3. Primary Outcome

Based on our pilot study, we had expected to observe significant differences between the groups, with an average reduction of 18 points in the IHM group and 9 points in the control group [23,24]. However, only 77 participants could be recruited for the study. Nevertheless, the total PSQ score of the IHM group for the ITT population decreased by an average of 25.25 points (95% CI [−30.77, −19.72]), indicating a 41% reduction. This demonstrates a substantial effect size (t(37) = −9.26, *p* < 0.001, d = −1.55). Notably, the mean score of 36.86 after six months of intervention is below the normal clinical threshold of 41.5. In contrast, the average score of the control group decreased by 5.41 (95% CI [−9.43, −1.39]), representing a 9% reduction. However, this reduction had only a small effect, and the mean score remains clinically abnormal (t(38) = −2.72, *p* = 0.010, d = −0.45). Although both groups showed a significant reduction in their total PSQ scores, the intervention group experienced a notably larger effect size compared to the control group (CG) (t(67.49) = −5.96, *p* < 0.001, d = −1.38), with an average reduction of −19.84 (95% CI [−26.47, −13.20]).

To validate the results further, we performed the same analysis with the per-protocol (PP) population. The effect within the IHM group (mean change = −24.46) and the difference between groups (mean difference = −19.42) were similar to those observed in the ITT population (t(30) = −8.15, *p* < 0.001, d = −1.70; t(52.26) = −5.43, *p* < 0.001, d = −1.37, respectively).

To fully assess the impact of the intervention over a 12-month period, we conducted a within-group comparison within the IHM group. Remarkably, the intervention demonstrated sustained efficacy, resulting in significant improvements by month 12 (t(37) = −10.33, *p* < 0.001, d = −1.70). On average, the PSQ total score decreased by 41% at month 6 and by 45% at month 12. Similar results were observed in the PP population (see Table 2). Overall, the IHM group derived substantial benefit from the intervention, whereas the control group exhibited the natural course of the disease, which did not reach clinical significance. Figure 3 shows the difference in PSQ total score at each time point.

### 3.4. PSQ Subscales

When examining the mean differences for the ITT population between the IHM group and the CG on the subscales at six months, significant differences with large effect sizes were observed for all sub-dimensions: worries (*p* < 0.001, d = −1.04); demands (*p* < 0.001, d = −0.88); joy (*p* < 0.001, d = 1.02); and tension (*p* < 0.001, d = −1.26). Furthermore, within-group analyses revealed that the IHM group exhibited substantial improvements in all dimensions after six and 12 months. In contrast, the CG showed significant changes only for the joy and tension sub-dimensions, with small effect sizes. See Table 2 for a more detailed overview of the statistical results.

### 3.5. Analysis of Covariance

To investigate the influence of the intervention on group variance while controlling for covariates, an ANCOVA was conducted using the intention-to-treat (ITT) population. The intervention accounted for 38% of the variance between the two groups and was found to have a significant effect on perceived stress scores (b = −21.02, *p* < 0.001). Although gender had no significant impact on changes in PSQ scores, baseline values had a modest significant effect (b = −7.16, *p* = 0.104; b = −0.35, *p* = 0.008, respectively).

Using the per-protocol (PP) population, the intervention’s effect was slightly lower, accounting for 34% of the variance (b = −20.51, *p* < 0.001). Additionally, neither baseline PSQ scores nor gender had a significant impact on variance (b = −0.28, *p* = 0.055; b = −5.79, *p* = 0.237, respectively). Table 3 provides a summary of the results.

### 3.6. Cross-Over Analysis

Due to the small sample size, an additional single-group crossover trial was conducted. In this study, we compared the original control group (who were previously on a waiting list) with the same participants after they had received the intervention (the new IHM group). We applied the same statistical analyses to the primary outcome as in the RCT described above, adjusting for the crossover design by modelling participant ID as a random effect in the ANCOVA and using paired *t*-tests.

The results of the crossover analysis were slightly less robust but consistent with those of the RCT. Using the ITT population, we observed an 8% reduction in the PSQ total score for the CG at six months (mean change = −5.05, 95% CI [−8.99, −1.10]). After the intervention, the same group’s PSQ total score decreased by a further 33% (mean change = −18.74, 95% CI [−15.95, −6.11]). While both reductions were statistically significant, the effect size was small during the waiting period and substantial after the intervention (CG: d = −0.44, *p* = 0.014; IHM: d = −0.99, *p* < 0.001). When we compared the mean difference between the groups at six months, we observed a significant improvement in the IHM group characterised by a large effect size (d = −0.88, *p* = 0.001). Notably, scores returned to clinically normal levels on average after six months of intervention, mirroring the experience observed in the IHM group in the RCT. The result for the per-protocol population was comparable to that in the imputed data. (See Table 4). Appendix A show the comparison of the mean differences from baseline at all time points as well as the mean PSQ total scores for all groups at all time points.

### 3.7. Secondary Outcomes

The IHM group showed significant improvements compared to the CG on all secondary outcomes except for pessimism after six months (see Table 5 for total scores and the Appendix A for a detailed overview of sub-dimensions). Within-group analyses of the IHM intervention, comparing scores at 12 months, consistently demonstrated significant improvements in all mental and psychological factors. These effects were more pronounced than those observed at six months. Table 5 and Appendix A summarise the main statistical findings. Appendix A provide a more detailed breakdown of the sub-dimensions. The statistical results are presented below, structured by psychological domain.

### 3.8. Stress Profile

Significant improvements were observed between the groups after six months of the intervention, favouring the IHM group. Large effect sizes were found for all burnout sub-dimensions, as well as for the total score (total score: d = −1.34, *p* < 0.001; discouragement: d = −1.22, *p* < 0.001; exhaustion: d = −0.99, *p* < 0.001; loss of motivation: d = −1.26, *p* < 0.001; see Table 5 and Appendix A).

Psycho-vegetative complaints also significantly improved, with a large effect size (total score: d = −1.24, *p* < 0.001). After six months, moderate between-group differences emerged in the sleep (d = −0.64, *p* = 0.010) and dizziness (d = −0.72, *p* = 0.004) sub-dimensions. Notably, the IHM group exhibited greater improvement in sleep quality (d = −1.16, *p* < 0.001) and dizziness (d = −0.91, *p* < 0.001), particularly after 12 months. This suggests an increasing difference between the groups over time (see Table 5 and Appendix A).

### 3.9. Mental Stress and Burden

Remission of mental load between 0 and 6 months was particularly evident in the significance values of the anxiety and depression sub-dimensions of the ISR (ICD Symptom Rating), with medium-to-high effect sizes (d = −0.79, *p* = 0.002; d = −0.76, *p* = 0.001, respectively). The total score showed a significant mean reduction of 0.32 between groups (d = −0.76, *p* = 0.001; see Appendix A).

### 3.10. Psychological Resource Profile

Similar positive results were found for the WHO-5 Well-Being Index, which is an early indicator of depression (d = 1.28, *p* < 0.001), as well as for vitality (d = 1.11, *p* < 0.001), self-efficacy (d = 0.78, *p* = 0.002) and optimism (d = 0.81, *p* = 0.002) (SWOP). There was no significant difference between groups for pessimism after six months (*p* = 0.386). However, pessimism improved considerably for the IHM group after 12 months (d = 1.78, *p* < 0.001). Significant differences in sense of coherence were observed between the groups, favouring the IHM group for the total score as well as for all subdimensions (total score: d = 1.07, *p* < 0.001; meaningfulness: d = 0.88, *p* < 0.001; comprehensibility: d = 0.83, *p* = 0.005; manageability: d = 0.64, *p* = 0.016). The IHM group also showed significant improvements in overall life satisfaction (d = 0.81, *p* = 0.001), particularly in the areas of friends/acquaintances (d = 0.73, *p* = 0.004), leisure activities/hobbies (d = 0.74, *p* = 0.003) and health (d = 0.78, *p* = 0.004). For an overview of all results, see Appendix A.

### 3.11. General Mood State Severity (VAS)

The general mood state severity index also showed large, significant improvements for the IHM group compared to the CG (d = −1.58, *p* < 0.001; see Table 4).

### 3.12. Adverse Events

Participants who dropped out before the end of the third month are not included in the list. This leaves 35 participants in the IHM group and 37 in the CG group. A total of 26 adverse events (AEs) were reported: 11 in the IHM group and 15 in the CG group. None were serious. Twenty-two of the 26 AEs were not related to the study and four were classified as being completely unrelated.

## 4. Discussion

Stress-related illnesses and disorders are prevalent in the general population and in primary healthcare settings, resulting in high rates of long-term sick leave [33,34]. The primary objective of the present study was to demonstrate the effectiveness of our programme in reducing perceived stress levels six months after the intervention, compared to those in the control waiting list group. This was measured using the Perceived Stress Questionnaire (PSQ), a widely used self-reporting tool for assessing perceived stress [25]. Six months after the start of the intervention, significant reductions in perceived stress levels were achieved, with an effect size of d = 1.38, as measured by the total PSQ score (M6) compared with the baseline value (M0). Although both groups showed reduced PSQ scores after six months, the effect was significantly and clinically more pronounced in the IHM group. Furthermore, group assignment alone explained 34–38% of the variance in the difference in total PSQ scores. In contrast, baseline scores ranging from 41.67 to 81.67 only had a medium effect on variance (10%), which decreased further to become non-significant when the per-protocol (PP) population was used instead of the intent-to-treat (ITT) population. These findings suggest that the intervention is appropriate for a wider range of patients with varying stress levels. Moreover, the lack of significant gender differences in the variances suggests that the uneven gender distribution between the groups did not substantially influence the results. However, the potential impact of gender and baseline stress levels on the observed effects requires further investigation. Sensitivity analyses and the crossover analysis confirmed the primary hypothesis, demonstrating a significant difference between groups regardless of the method employed. In all cases, mean perceived stress (PSQ) scores in the IHM group decreased significantly, reaching clinically normal levels (≤41.5) within six months. Remarkably, these low levels were maintained for up to 12 months, with further improvements observed at six and 12 months (−41% and −45%, respectively).

The PSQ-Score outcome was broadly consistent with that of an internet-delivered, therapist-supported online cognitive behavioural therapy (CBT) treatment, which Lindsäter et al. [35] compared with a waiting list group. After six months, a strong effect size was observed in the PSQ, albeit smaller than in our study: Cohen’s d = 1.09. A recent study by Sennerstam et al. [36] compared internet-delivered CBT with a lifestyle programme, finding no clinical differences in reducing perceived stress symptoms (Cohen’s d = 1.19 vs. 1.06). These findings support our own and indicate that lifestyle-based interventions can be as effective as internet-delivered CBT in reducing perceived stress.

According to Lindsäter et al. [35] and Sennerstam et al. [36], patients with adjustment disorders (AD) in the International Classification of Diseases (ICD-11) typically experience symptoms such as persistent worry, sleep disturbances, and somatoform complaints, while those with exhaustion disorders (ED) are more likely to suffer from cognitive impairment and pronounced fatigue. Although these categories differ in symptom emphasis, considerable overlap exists across mental health disorders. This suggests that interventions targeting stress-related symptoms, such as the present lifestyle-based, blended learning approach, may be applicable to these diagnostic groups more broadly.

Psychological distress and long-term stress exposure are progressive health problems that have also been linked to an unhealthy lifestyle, decreased quality of life and reduced well-being. Such lifestyle factors include physical activity and sedentary behaviour [7,37,38], dietary patterns and body mass index (BMI) [39,40], sleep problems [41] and smoking [42]. All of these factors may influence stress, depression and anxiety. The increase in mental health issues requires the consistent promotion of resilience and stress management, as well as lifestyle-based prevention programmes. Our IHM-Lifestyle Programme takes a salutogenic approach to encourage a positive attitude in patients, helping them to cope with everyday stressors and to promote their quality of life. The programme also helps patients to manage their attitudes, behaviour, symptoms and disease.

Furthermore, our secondary analyses revealed that our comprehensive lifestyle intervention reduced stress levels and improved various aspects of mental and physical health. This included a significant reduction in symptoms of mental distress, such as depression and anxiety. There were also improvements in functional somatic symptoms and neurovegetative functions, such as orthostatic dizziness, sleep regulation, and overall mood severity. The study also demonstrated that psychological resources, including well-being, vitality, life satisfaction, sense of coherence, self-efficacy, and optimism, were advantageous. Large within-group effect sizes were observed for most of these indicators from pre- to post-treatment, as well as significant between-group results compared with the waiting list group. These results were maintained at the 1-year follow-up.

A systematic review and meta-analysis by Amiri et al. [14] examined the effect of lifestyle interventions on depression, anxiety, and stress in randomised controlled trials. The findings revealed that adopting a healthy lifestyle improves mental health and significantly reduces stress, with pronounced effects in patients with depressive symptoms and in women. Changes to lifestyle, such as increased exercise, a healthier diet and improved sleep quality, can effectively reduce stress by lowering cortisol levels and increasing endorphins [43,44]. Furthermore, the findings suggest that psychological factors, such as increased mindfulness and improved sleep quality, may have a positive impact. This is particularly relevant given that sleep disturbances are closely linked to cognitive impairments, such as memory lapses and attention deficits, which may act as mediators between insomnia and emotional disorders, such as anxiety and depression [45].

However, the effectiveness of such interventions depends not only on the basic methodology, but also on how they are implemented. Lifestyle interventions can take the form of personal guidance during seminars and courses, personal training or internet-based training, either alone or in combination. In this context, Brinsley et al.’s [46] meta-analysis and review of digital lifestyle studies is interesting. The primary outcome was the change in symptoms of mental disorders from pre- to post-intervention. Compared to the results of our study, the authors found small to moderate effects for depression, small effects for anxiety and stress, and no effects on well-being.

The IHM programme uses a combination of teaching methods, such as classroom teaching supplemented by personal online support, as part of a blended learning approach. This approach may be more effective than face-to-face or online instruction alone. However, previous research [47,48,49] has demonstrated that, while internet-based interventions provide short-term stress relief, they do not lead to long-term benefits. However, longer interventions involving refresher seminars require greater commitment from participants, which likely contributed to the higher dropout rate observed in our study. This emphasises the importance of balancing intervention intensity with participant engagement to ensure both effectiveness and adherence.

### Strengths and Limitations

One strength of our study is that we recruited a whole cohort of insured individuals from a statutory health insurance fund that covers a large area around a clinic. There were no prior restrictions based on symptoms or conditions. Although we were able to reach many people, the outbreak of the pandemic made it difficult, and sometimes impossible, to recruit participants for face-to-face training groups. Consequently, the randomisation process did not involve the desired number of participants, resulting in an imbalanced gender distribution. Therefore, gender was included as a covariate in the ANCOVA model and was not a significant predictor of the outcome. Although statistically and clinically relevant effects were measured despite the small number of cases in the study, there is still a risk that the study was underpowered and that the actual effect of the program is overestimated.

## 5. Conclusions

IHM is an effective intervention for reducing stress and improving other aspects of mental health, with positive effects lasting up to a year. Further research is recommended to investigate the parameters of the intervention and its long-term effects. Finally, it is important to consider how best to integrate lifestyle programmes into service delivery.

## Figures and Tables

**Figure 1 healthcare-13-03181-f001:**
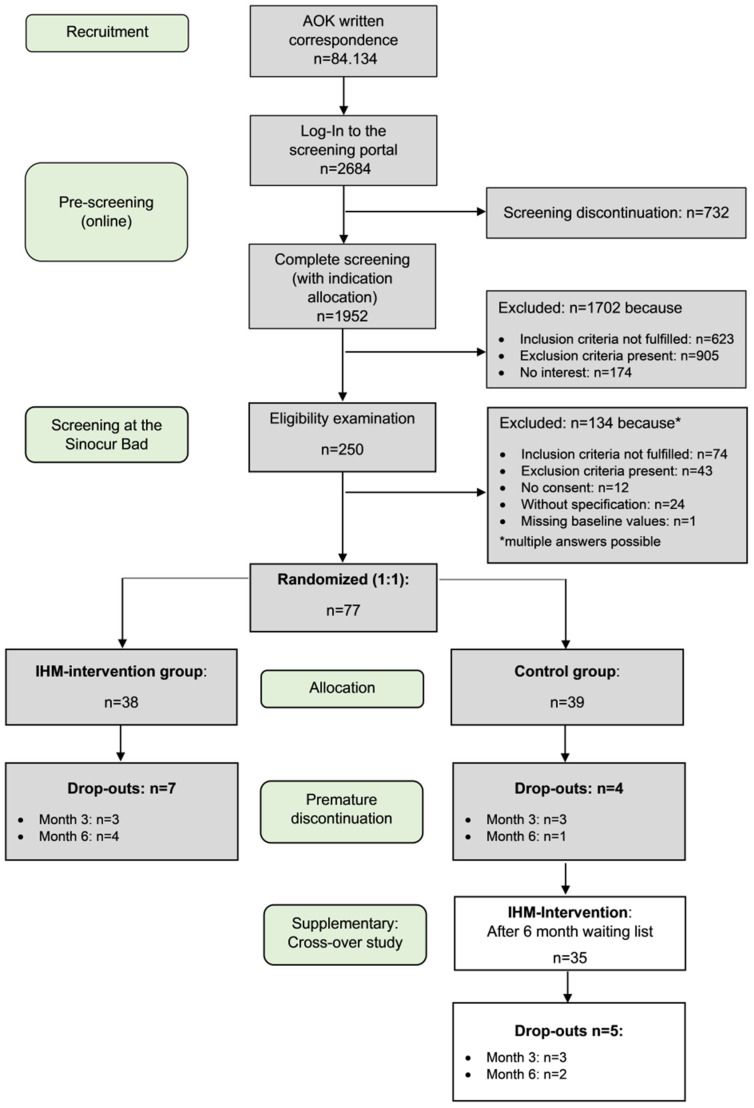
CONSORT study flowchart: Due to COVID-19, data for month 3 is missing in two cases. However, these participants did not drop out of the study and were not excluded as the reason for the missing data was not related to the study.

**Figure 2 healthcare-13-03181-f002:**
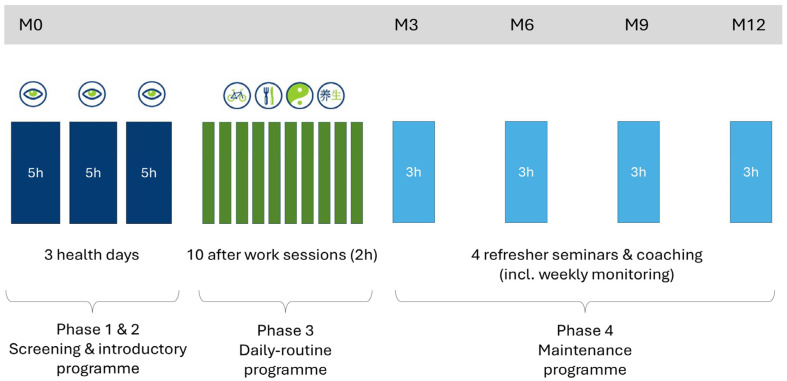
Schematic course of the IHM: Illustration of the structure and timeline of the Individual Health Management (IHM) intervention across its phases. (
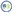
 = Yang Sheng).

**Figure 3 healthcare-13-03181-f003:**
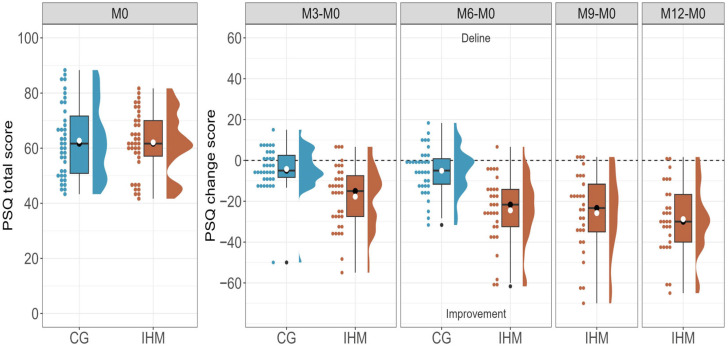
Perceived Stress: Distribution of mean differences from baseline (M0) after three months (M3), six months (M6), nine months (M9) and twelve months (M12) per group (CG = control group, IHM = intervention group; no imputation). The dashed line indicates no change. Values above the line show a decline and values below the line an improvement. White circles represent the mean, black lines and circles the median.

**Table 1 healthcare-13-03181-t001:** Baseline demographic and clinical characteristics.

	Intervention Group (n = 38)		Control Group (n = 39)		Between Group Differences	In Total (n = 77)	
	Mean	SD	Mean	SD	*p*	Mean	SD
Age (years)	44.32	10.05	45.18	11.35	0.725	44.75	10.67
	**n**	**%**	**n**	**%**	** *p* **	**n**	**%**
Female	29	76.3	35	89.7	<0.001 ***	64	83.1
Male	9	23.7	4	10.3		13	16.9
School education					0.529		
No school-leaving certificate	0	0.0	1	2.6		1	1.3
Secondary general school certificate	15	40.5	11	28.2		26	34.2
Intermediate school certificate	15	40.5	19	48.7		34	44.7
University entrance qualification	6	16.2	8	20.5		14	18.4
Other degree	1	2.7	0	0.0		1	1.3
Vocational education					0.207		
No completed vocational training	4	10.8	4	10.3		8	10.5
Completed apprenticeship	23	62.2	18	46.2		41	54.0
Vocational School	7	18.9	11	28.2		18	23.7
University of applied science	1	2.7	3	7.7		4	5.3
University degree	0	0.0	2	5.1		2	2.6
Other degree	2	5.4	1	2.6		3	3.9
Employment status					0.748		
Employed	34	91.9	35	89.7		69	90.8
Temporarily retired	1	2.7	0	0.0		1	1.3
Retired	0	0.0	2	5.1		2	2.6
Employed without remuneration	1	2.7	1	2.6		2	2.6
Without specification	1	2.7	1	2.6		2	2.6
Living condition					0.286		
Individual household	5	13.5	9	23.1		14	18.4
Multi-person household	32	86.5	30	76.9		62	81.6
Smoking					0.720		
No	35	92.1	35	89.7		70	90.9
Yes	3	7.9	4	10.3		7	9.1
Alcohol					0.417		
No	36	94.7	35	89.7		71	92.2
Yes	2	5.3	4	10.3		6	7.8
PSQ	**Mean**	**SD**	**Mean**	**SD**	** *p* **	**Mean**	**SD**
Total score	62.11	11.30	62.74	13.19	0.822	62.42	12.22
Worries	51.23	18.61	51.97	17.58	0.859	51.60	17.98
Demands	64.03	22.49	57.78	18.88	0.191	60.87	20.84
Joy	36.67	14.52	28.03	15.83	0.015 *	32.29	15.71
Tension	69.82	14.35	69.23	18.14	0.874	69.52	16.27
Secondary outcomes							
Tedium-Measure Burnout	4.19	0.54	4.33	0.54	0.266	4.26	0.54
Psycho-vegetative complaints	56.74	13.86	53.72	13.12	0.330	55.21	13.48
ISR total scores	1.13	0.51	1.07	0.42	0.560	1.10	0.47
General mood state severity (VAS)	58.82	15.24	53.69	15.12	0.143	56.22	15.30
Well-being (WHO-5)	7.89	3.65	10.56	10.48	0.140	9.25	7.95
Vitality (SF-36)	35.00	15.55	37.69	15.34	0.447	36.36	15.40
Self-efficacy (SWOP)	12.21	2.73	11.41	2.45	0.180	11.81	2.61
Optimism (SWOP)	5.32	1.36	4.74	1.25	0.058	5.03	1.33
Pessimism (SWOP)	5.00	1.19	5.00	1.19	1	5.00	1.18
Sense of Coherence (SOC)	54.53	9.16	54.54	9.44	0.995	54.53	9.25
Life satisfaction (FLZ)	33.34	27.37	29.97	26.70	0.587	31.64	26.91

Note. Table 1 shows the sociodemographic characteristics of the intervention (IHM) and control (CG) groups, including means and standard deviations (SD), *p*-values (significance: *p* < 0.05 *, *p* < 0.001 ***) from independent between-group comparisons using either a *t*-test or a Wilcoxon-rank-sum test for categorial parameters, and overall means and SD for the total sample. Please note that data for one participant (IHM) is missing for education, employment details and living conditions due to a technical error.

**Table 2 healthcare-13-03181-t002:** Findings from the statistical analysis of the PSQ, broken down by time point and group.

	Intervention Group (n = 38)				Control Group (n = 39)				Between Groups		ANCOVA
	Mean	SD	ΔM0	Cohen’s d	Mean	SD	ΔM0	Cohen’s d	Mean Difference ΔM0 [95% CI]	Cohen’s d	Parameter Estimate for Group Factor [95% CI]
PSQ total score ITT											
M0	62.11	11.30			62.74	13.19			−0.63 [−6.21; 4.95]		
M3	44.71	14.95	−17.39	−1.13 ***	59.06	15.88	−3.68	−0.33.	−13.72 [−20.14; −7.30]	−1.02 ***	−14.70 [−20.94; −8.45]
M6	36.86	15.46	−25.25	−1.55 ***	57.32	15.60	−5.41	−0.45 **	−19.84 [−26.47; −13.20]	−1.38 ***	−21.02 [−27.34; −14.70]
M9	36.84	15.71	−25.27	−1.37 ***							
M12	33.90	14.43	−28.20	−1.70 ***							
Sensitivity analysis for PSQ total score PP											
M3	44.76	15.44	−17.71	−1.31 ***	59.19	16.43	−4.24	−0.27 *	−13.48 [−19.84; −7.11]	−1.01 ***	−14.20 [−20.54; −7.85]
M6	37.15	16.70	−24.46	−1.70 ***	57.57	16.06	−5.05	−0.34 *	−19.42 [−26.10; −12.43]	−1.37 ***	−20.51 [−27.48; −13.54]
M9	36.04	17.39	−25.80	−1.78 ***							
M12	33.91	15.88	−28.79	−2.13 ***							
PSQ worries ITT											
M0	51.23	18.61			51.97	17.58			−0.74 [−8.96; 7.48]		
M3	34.74	14.97	−16.49	−0.92 ***	48.87	19.68	−3.09	−0.18	−13.40 [−21.80; −4.99]	−0.76 **	−14.36 [−21.87; −6.84]
M6	27.19	16.56	−24.04	−1.18 ***	47.78	20.45	−4.19	−0.24	−19.85 [−28.85; −10.85]	−1.04 ***	−21.48 [−29.61; −13.36]
M9	25.75	17.49	−25.47	−1.24 ***							
M12	24.30	15.53	−26.93	−1.38 ***							
PSQ demands ITT											
M0	64.04	22.49			57.78	18.88			6.26 [−3.16; 15.68]		
M3	45.79	22.04	−18.25	−0.87 ***	56.48	23.03	−1.30	−0.08	−16.95 [−26.56; −7.33]	−0.89 ***	−15.96 [−25.19; −6.73]
M6	42.60	23.60	−21.44	−1.09 ***	52.91	20.87	−4.87	−0.27	−16.57 [−25.82; −7.31]	−0.88 ***	−15.33 [−24.47; −6.19]
M9	45.00	23.95	−19.04	−0.79 ***							
M12	39.30	20.81	−24.74	−1.30 ***							
PSQ joy ITT											
M0	36.67	14.52			28.03	15.83			8.63 [1.73; 15.54]		
M3	50.79	19.46	14.12	0.78 ***	35.37	17.21	7.33	0.52 **	6.79 [−1.18; 14.76]	0.42	10.16 [2.19; 18.14]
M6	60.18	19.83	23.51	1.28 ***	33.90	18.91	5.86	0.37 *	17.65 [9.07; 26.22]	1.02 ***	21.51 [12.58; 30.44]
M9	61.70	20.19	25.04	1.13 ***							
M12	62.07	20.71	25.40	1.18 ***							
PSQ tension ITT											
M0	69.82	14.35			69.23	18.14			0.59 [−6.85; 8.03]		
M3	49.11	17.99	−20.72	−0.96 ***	66.26	20.59	−2.97	−0.18	−17.75 [−26.97; −8.52]	−0.91 ***	−18.48 [−26.92; −10.03]
M6	37.82	18.27	−32.00	−1.39 ***	62.51	22.64	−6.72	−0.41 *	−25.28 [−34.91; −15.66]	−1.26 ***	−25.87 [−35.04; −16.70]
M9	38.30	21.68	−31.53	−1.24 ***							
M12	34.09	18.17	−35.74	−1.56 ***							

Note: The table shows *t*-test results comparing the IHM and CG, as well as paired *t*-tests assessing changes over time. For each time point, mean scores (±SD), mean change scores from baseline (ΔM0), effect sizes (Cohen’s d: <0.05 small, 0.5–0.8 medium, >0.8 large) and *p*-values (Significance: *p* < 0.05 *, *p* < 0.01 **, *p* < 0.001 ***) are reported. Between-group mean differences and 95% confidence intervals (CIs) are presented, along with the parameter estimate for the group factor from the ANCOVA model, adjusted for baseline scores and gender. SD values and Cohen’s d for the intent-to-treat population (ITT) were calculated from multiply imputed data as the mean value across imputations. A sensitivity analysis based on the PP sample is also included. The sample size for each month is as follows: IHM: M3 n = 35, M6 n = 31, M9 n = 29, M12 n = 29; PP: M3 n = 35, M6 n = 35.

**Table 3 healthcare-13-03181-t003:** ANCOVA results for the differences between month 6 and month 0 adjusted for gender and baseline values.

ANCOVA Results	Estimate	SE	df	[95% CI]	t	*p*	Par. n^2^
**ITT Population**							
Intercept	30.37	11.29	61.81	[7.81; 52.93]	2.69	0.009	
Group (IHM/CC)	−21.02	3.17	68.04	[−27.34; −14.70]	−6.63	<0.001	0.38
PSQ baseline score	−0.35	0.13	67.74	[−0.61; −0.10]	−2.73	0.008	0.10
Gender	−7.16	4.34	63.44	[−15.84; 1.52]	−1.65	0.104	0.04
**PP Population**							
Intercept	17.73	10.37	62	[−2.99; 38.45]	1.71	0.092	
Group (IHM/CC)	−20.51	3.49	62	[−27.47; −13.54]	−5.89	<0.001	0.34
PSQ baseline score	−0.28	0.14	62	[−0.56; 0.01]	−1.96	0.055	0.06
Gender	−5.79	4.85	62	[−15.48; 3.90]	−1.20	0.237	0.02

Note: the table shows the estimated marginal means (±standard error and 95% confidence interval (CI)) for the intent-to-treat (ITT) and per-protocol (PP) populations, alongside the ANCOVA model results including the intercept, group (control group (CG) vs. intervention group (IHM)), perceived stress baseline score (PSQ) and gender as predictors. The associated *p*-value for each variable in the model indicates statistical significance, and partial eta squared (par. η^2^) reflects the effect size by representing the proportion of variance explained by that variable after accounting for other factors. For the ITT population, partial n^2^ was calculated from multiply imputed data as the mean value across imputations.

**Table 4 healthcare-13-03181-t004:** Cross-over: Statistical results for the comparison between and within groups.

	Intervention Group (n = 35)				Control Group (n = 35)				Between Groups		ANCOVA
	Mean	SD	ΔM0	Cohen’s d	Mean	SD	ΔM0	Cohen’s d	Mean Difference ΔM0 [95% CI]	Cohen’s d	Parameter Est. for Group [95% CI]
PSQ total score ITT											
M0	57.57	16.06			62.62	13.17					
M3	46.54	20.72	−11.03	−0.80 ***	58.37	15.92	−4.25	−0.39 *	−6.78 [−12.42; −1.14]	−0.54 *	−7.64 [−12.94; −2.34]
M6	38.83	22.23	−18.74	−0.99 ***	57.57	16.06	−5.05	−0.44 *	−13.69 [−21.74; −5.64]	−0.88 **	−15.23 [−22.36; −8.10]
M9	45.68	21.70	−11.89	−0.69 ***							
M12	35.80	18.58	−21.77	−1.57 ***							
Sensitivity analysis for PSQ total score PP											
M3	46.24	20.87	−11.61	−0.59 ***	58.38	15.96	−4.41	−0.29 *	−6.67 [−12.10; −1.23]	−0.55 *	−7.72 [−12.83; −2.60]
M6	38.33	22.08	−19.11	−0.96 ***	57.57	16.06	−5.05	−0.34 *	−13.78 [−22.04; −5.51]	−0.89 **	−15.56 [−22.75; −8.36]
M9	44.81	23.35	−11.16	−0.52 **							
M12	34.47	18.07	−21.60	−1.22 ***							

Note: this table summarises the results of dependent samples *t*-tests, which compare the new intervention group (IHM) with the control group (CG), and paired samples *t*-tests, which assess changes within groups over time. For each time point, the following are reported: mean scores (±standard deviation, SD); mean change scores from baseline (ΔM0); effect sizes (Cohen’s d: <0.05 small effect, 0.5–0.8 medium effect, >0.8 large effect); and *p*-values (significance: *p* < 0.05 *, *p* < 0.01 **, *p* < 0.001 ***). Between-group mean differences with 95% confidence intervals (CIs) are presented, along with the parameter estimate for the group factor from the ANCOVA model, adjusted for baseline scores and gender. SD values and Cohen’s d for the intent-to-treat (ITT) population were calculated as the mean value across multiply imputed data. A sensitivity analysis based on the per protocol (PP) sample is also included. The sample sizes for the PP for each month are as follows: IHM: n = 35 for M0, n = 31 for M3, n = 30 for M6, n = 26 for M9, n = 25 for M12; CC: n = 34 for M3, n = 35 for M6 (data missing for one person between time points due to COVID-19).

**Table 5 healthcare-13-03181-t005:** Results for the main secondary outcomes.

	Intervention Group (n = 38)				Control Group (n = 39)				Between Groups		ANCOVA
	Mean	SD	ΔM0	Cohen’s d	Mean	SD	ΔM0	Cohen’s d	Mean Difference ΔM0 [95% CI]	Cohen’s d	Parameter Est. for Group [95% CI]
Tedium-Measure (Burnout): total score											
M0	4.19	0.54			4.33	0.54					
M3	3.42	0.75	−0.77	−1.15 ***	4.19	0.71	−0.14	−0.25	−0.63 [−0.92; −0.35]	−1.03 ***	−0.69 [−0.98, −0.40]
M6	2.99	0.71	−1.20	−1.50 ***	4.06	0.70	−0.27	−0.46 **	−0.94 [−1.26; −0.61]	−1.34 ***	−1.06 [−1.36, −0.75]
M9	2.75	0.68	−1.45	−1.71 ***							
M12	2.74	0.73	−1.45	−1.77 ***							
Psycho-vegetative complaints total score											
M0	56.74	13.86			53.72	13.12					
M3	41.92	14.01	−14.82	−1.03 ***	51.98	14.86	−1.74	−0.19	−13.08 [−18.81; −7.35]	−1.08 ***	−13.03 [−18.43, −7.63]
M6	35.58	15.43	−21.15	−1.24 ***	50.77	15.08	−2.94	−0.25	−18.21 [−25.06; −11.36]	−1.24 ***	−17.99 [−24.34, −11.64]
M9	34.41	14.12	−22.33	−1.25 ***							
M12	31.07	14.45	−25.67	−1.48 ***							
ISR: total score											
M0	1.13	0.51			1.07	0.42					
M3	0.87	0.47	−0.26	−0.66 ***	1.07	0.47	0.00	0.01	−0.27 [−0.44, −0.10]	−0.72 **	−0.29 [−0.44, −0.14]
M6	0.72	0.43	−0.41	−0.85 ***	0.98	0.37	−0.09	−0.25	−0.32 [−0.52, −0.12]	−0.76 **	−0.33 [−0.48, −0.17]
M9	0.66	0.41	−0.47	−0.99 ***							
M12	0.53	0.36	−0.61	−1.21 ***							
WHO well-being index											
M0	7.89	3.65			9.26	4.73					
M3	12.86	5.28	4.97	0.95 ***	9.96	5.34	0.71	0.16	4.26 [1.94, 6.58]	0.86 ***	3.85 [1.59, 6.11]
M6	15.03	4.28	7.14	1.42 ***	9.67	5.18	0.41	0.08	6.73 [4.31, 9.15]	1.28 ***	6.05 [3.88, 8.22]
M9	14.61	5.92	6.71	1.13 ***							
M12	15.82	4.25	7.92	1.65 ***							
Vitality											
M0	35.00	15.55			37.69	15.34					
M3	52.08	20.54	17.08	0.82 ***	40.31	20.59	2.62	0.17	14.46 [5.34, 23.59]	0.79 **	14.79 [5.96, 23.62]
M6	60.58	19.37	25.58	1.06 ***	40.17	19.06	2.47	0.15	23.10 [4.88, 13.35]	1.11 ***	22.49 [13.66, 31.32]
M9	60.05	22.37	25.05	1.03 ***							
M12	63.55	18.44	28.55	1.17 ***							
Self-efficacy (SWOP)											
M0	12.21	2.73			11.41	2.45					
M3	13.26	2.70	1.05	0.47 **	12.12	2.33	0.71	0.37	0.34 [−0.73, 1.41]	0.16	0.70 [−0.34, 1.75]
M6	14.59	2.53	2.38	0.89 ***	11.83	2.28	0.42	0.19	1.96 [0.74, 3.17]	0.78 **	2.49 [1.41, 3.56]
M9	15.04	2.88	2.83	1.10 ***							
M12	15.47	2.44	3.26	1.23 ***							
Sense of Coherence (SOC): total score											
M0	54.53	9.16			54.54	9.44					
M3	60.10	11.74	5.57	0.51 **	55.22	11.06	0.68	0.08	4.89 [0.25, 9.53]	0.50 *	5.36 [0.85, 9.89]
M6	64.61	11.59	10.08	1.01 ***	54.76	11.18	0.22	0.02	9.87 [5.00, 14.73]	1.07 ***	10.04 [5.17, 14.91]
M9	65.90	10.61	11.38	1.03 ***							
M12	65.23	11.16	10.71	0.90 ***							
Life satisfaction (FLZ): total score											
M0	33.34	27.37			29.97	26.70					
M3	42.51	25.83	9.17	0.33.	40.24	26.70	10.26	0.50 **	−1.09 [13.32, 11.14]	−0.04	1.00 [−9.84, 11.84]
M6	59.90	28.28	26.56	0.75 ***	33.13	26.40	3.16	0.16	23.40 [9.63, 37.16]	0.81 **	27.53 [15.77, 39.30]
M9	63.50	29.29	30.16	0.91 ***							
M12	62.93	32.75	29.59	0.83 ***							
General mood state severity (VAS)											
M0	58.82	15.24			53.69	15.12					
M3	36.70	16.16	−22.11	−1.18 ***	51.33	20.87	−2.37	−0.11	−19.75 [−29.29, −10.20]	−0.96 ***	−17.45 [−26.08, −8.81]
M6	29.88	14.92	−28.93	−1.45 ***	57.64	17.16	3.94	0.18	−32.88 [−42.76, −22.99]	−1.58 ***	−29.71 [−37.74, −21.68]
M9	30.86	18.20	−27.96	−1.20 ***							
M12	28.50	15.63	−30.32	−1.33 ***							

Note: This table summarises the results of independent samples *t*-tests comparing the intervention group (IHM) and the control group (CG), as well as the results of paired samples *t*-tests assessing changes within each group over time. For each time point, the following are reported: mean scores (± standard deviation, SD); mean change scores from baseline (ΔM0); effect sizes (Cohen’s d: <0.05 small effect, 0.5–0.8 medium effect, >0.8 large effect); and *p*-values (significance: *p* < 0.05 *, *p* < 0.01 **, *p* < 0.001 ***). Between-group mean differences with 95% confidence intervals (CIs) are presented, along with the parameter estimate for the group factor from the ANCOVA model, adjusted for baseline scores and gender. All samples are based on the intent-to-treat (ITT) population. SD values and Cohen’s d were calculated as the mean value across multiple inputs.

## Data Availability

The original contributions presented in this study are included in the article/Appendix A. Further inquiries can be directed to the corresponding author. The statistical data from the study can be found in the final report to the project sponsor. This will be available as a supplement download as part of the TCM Clinic Report 2023/4, which is scheduled to be published in digital book form by Springer Verlag in mid-2026.

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
