# Peer review of "Individual Health Management (IHM) for Stress—A Randomised Controlled Trial (TALENT II Study)"

_healthcare, 2025, doi:10.3390/healthcare13233181_

Round 1
Reviewer 1 Report
Comments and Suggestions for Authors
General comments
This manuscript “Individual Health Management (IHM) For Stress – A Randomised Controlled Trial (TALENT II Study)” is a study that examines the effectiveness of an adapted lifestyle intervention in reducing stress. Overall, the paper is well-written, and the authors describe the Individual Health Management intervention, and the outcomes related to the study.
A few major comments include
i) a need to clearly explain the intervention
ii) lack of citations in the introduction
iii) clearly addressing how the small sample size, drop-outs impacted the power if the study.
Below are specific comments. I hope you will find them constructive and helpful.

Author Response
Please see my reply attached! Many thanks!

Reviewer 2 Report
Comments and Suggestions for Authors
The number of males in the control and treatment groups is 9 and 4, respectively. This is not an acceptable number. Even in animal studies, a number of 4 is unacceptable.
The number of participants should be closer to the number of female participants. The number of male participants must definitely be increased.
The human study here and the heterogeneity of the people involved are too great. Their psychological and physiological states are very heterogeneous.
Data analysis conducted in this manner does not provide reliable data.
Also, at what time were the stress-related measurements taken?
Why were not glucocorticoids measured in saliva or blood for stress?
Author Response
Please see my reply attached. Many thanks!

Reviewer 3 Report
Comments and Suggestions for Authors
The authors investigated the effectiveness of the Individual Health Management lifestyle programme, a combination of education, lifestyle modification, and online support, in reducing perceived stress in patients with stress-related disorders. The aim was to determine whether this programme was more effective than a waiting-list control group and whether it improved additional mental and physical health indicators. The study showed that the programme significantly reduced perceived stress levels, with a large effect that was maintained for up to one year after the intervention. In addition to reducing stress, the programme improved symptoms of depression, anxiety, functional somatic complaints, sleep, mood, and psychological resources such as self-efficacy and vitality. The programme proved to be as effective as internet-delivered cognitive behavioral therapy and broadly applicable to patients with varying levels of stress.
Minor comments:
Which specific elements of lifestyle interventions (physical activity, nutrition, sleep, mindfulness) contribute the most to stress reduction according to previous studies?
Is there any evidence of biological markers (cortisol, inflammatory markers, neuropeptides) that confirm the long-term effects of lifestyle interventions?
Have differences in the effectiveness of lifestyle interventions been found between clinical populations (e.g., anxiety disorders, depression, burnout)?
To what extent are the results of similar programs sustainable for more than one year, and what does the literature say about long-term follow-up?
Author Response
Please see my reply attached. Many thanks.

Round 2
Reviewer 2 Report
Comments and Suggestions for Authors
I congratulate the authors on their meticulous and beautiful work. However, the number of male subjects needs to increase. This is not acceptable. Best regards.
Author Response
Hello,
please see reply attached.
